# Ginsenoside Rb1 Attenuates High Glucose-Induced Oxidative Injury via the NAD-PARP-SIRT Axis in Rat Retinal Capillary Endothelial Cells

**DOI:** 10.3390/ijms20194936

**Published:** 2019-10-05

**Authors:** Chunlan Fan, Qing Ma, Meng Xu, Yuan Qiao, Yi Zhang, Pin Li, Yucong Bi, Minke Tang

**Affiliations:** 1School of Life Sciences, Beijing University of Chinese Medicine, Beijing 102488, China; fanchunlan77@163.com (C.F.); 20170941210@bucm.edu.cn (P.L.); byc1206@163.com (Y.B.); 2School of Chinese Materia Medica, Beijing University of Chinese Medicine, Beijing 102488, China; maqing988@163.com (Q.M.); xm_dreaming@163.com (M.X.); yizhang714@163.com (Y.Z.); 3Institute of Chinese Materia Medica, Shaanxi Provincial Academy of Traditional Chinese Medicine, Xi’an 710003, Shaanxi, China; qy0508@163.com

**Keywords:** Ginsenoside Rb1, high glucose, retinal capillary endothelial cells, NAD^+^, PARP, sirtuin

## Abstract

(1) Aims: The present study aimed to observe the effects of Ginsenoside Rb1 on high glucose-induced endothelial damage in rat retinal capillary endothelial cells (RCECs) and to investigate the underlying mechanism. (2) Methods: Cultured RCECs were treated with normal glucose (5.5 mM), high glucose (30 mM glucose), or high glucose plus Rb1 (20 μM). Cell viability, lactate dehydrogenase (LDH) levels, the mitochondrial DNA copy number, and the intracellular ROS content were measured to evaluate the cytotoxicity. Superoxide dismutase (SOD), catalase (CAT), nicotinamide adenine dinucleotide phosphate (NADPH) oxidase (NOX), poly(ADP-ribose) polymerase (PARP), and sirtuin (SIRT) activity was studied in cell extracts. Nicotinamide adenine dinucleotide (NAD^+^)/NADH, NADPH/NADP^+^, and glutathione (GSH)/GSSG levels were measured to evaluate the redox state. The expression of nicotinamide mononucleotide adenylyltransferase 1 (NMNAT1), SIRT1, and SIRT3 was also evaluated after Rb1 treatment. (3) Results: Treatment with Rb1 significantly increased the cell viability and mtDNA copy number, and inhibited ROS generation. Rb1 treatment increased the activity of SOD and CAT and reduced the activity of NOX and PARP. Moreover, Rb1 enhanced both SIRT activity and SIRT1/SIRT3 expression. Additionally, Rb1 was able to re-establish the cellular redox balance in RCECs. However, Rb1 showed no effect on NMNAT1 expression in RCECs exposed to high glucose. (4) Conclusion: Under high glucose conditions, decreases in the reducing power may be linked to DNA oxidative damage and apoptosis via activation of the NMNAT-NAD-PARP-SIRT axis. Rb1 provides an advantage during high glucose-induced cell damage by targeting the NAD-PARP-SIRT signaling pathway and modulating the redox state in RCECs.

## 1. Introduction

Diabetic retinopathy (DR) is one of the major complications of diabetes mellitus (DM) and is the leading cause of vision loss and blindness among working-age people in developed countries [1]. The underlying pathophysiological mechanisms of DR are still incompletely understood. Previous studies have found that retinal endothelial cell apoptosis is an early event in DR. In diabetic animals, exposure of the retinal vasculature to high glucose results in a loss of retinal capillary endothelial cells (RCECs) through apoptosis [2]. Additional studies using human RCECs [3], bovine RCECs [4], and rat RCECs [5] have reported an increase in apoptosis when cells are exposed to high glucose.

Hyperglycemia is the main cause of diabetes complications, including DR [6]. Oxidative stress is increased in diabetes, and the overproduction of reactive oxygen species (ROS) correlates with complications of diabetes [7]. Mitochondrial DNA (mtDNA) is susceptible to endogenous and exogenous oxidative damage, and mtDNA damage leads to mitochondrial dysfunction, tissue damage, insulin resistance, and other complications observed in DM [8,9]. Several investigators have shown that the hyperglycemic environment present in DM leads to enhanced ROS production, DNA damage, and a decrease in nicotinamide adenine dinucleotide (NAD^+^) levels, all of which have been linked to a decrease in sirtuins (SIRT) expression and involve the activation of poly(ADP-ribose) polymerase (PARP) [10,11,12]. Nicotinamide mononucleotide adenylyltransferase 1 (NMNAT1) is a nuclear enzyme in all organisms that is essential for NAD synthesis in the salvage/recycling pathway, which regulates the functions of NAD-dependent enzymes such as the protein deacetylase SIRT1 and PARP [13,14]. Recent studies have revealed that NMNAT1 appears to have neuroprotective properties in the retina, brain, and central nervous system, and mutations in NMNAT1 have been identified to cause a non-syndromic early form of blindness [15,16]. However, whether high glucose can induce NMNAT1 expression in RCECs remains unknown.

Nicotinamide adenine dinucleotide phosphate (NADPH) is a coenzyme widely involved in a broad range of oxidation reduction reactions that are important for the maintenance of reduced glutathione (GSH) levels. High glucose-induced activation of the polyol pathway, which is catalyzed by aldose reductase (AR), converts glucose to sorbitol with the concomitant oxidation of NADPH to NADP^+^ [17]. The depletion of intracellular NADPH and GSH promotes NADPH oxidase (NOX) activity, which has emerged as a major source of ROS in endothelial cells. Previous studies have shown that hyperglycemia leads to an alteration in the redox potential of cells, particularly the NADPH/NADP^+^ ratio and GSH levels [18].

*Panax notoginseng* (Burk.) *F.H. Chen* (Sanqi) has been extensively used in traditional oriental medicine for thousands of years. Although the components of the *Panax notoginseng* root vary across sources and species, Ginsenoside Rb1 (Rb1), Ginsenoside Rg1 (Rg1), and Notoginsenoside R1 (NR1) are the main active compounds [19]. Recent studies have shown that Ginsenoside Rb1 and Notoginsenoside R1 have diverse pharmacological properties, including antioxidative, anti-inflammatory, antidiabetic, cardioprotective, and neuroprotective properties, both in vitro and in vivo [20,21,22,23,24]. In our previous study, Notoginsenoside R1 was endowed with a significant protective function against high glucose-induced oxidative injury in RCECs by modulating the intracellular redox state [25]. Whether Rb1 is retinoprotective and has any effect on the activity of the NMNAT-NAD-PARP-SIRT axis under high glucose conditions is unknown. The present study aimed to evaluate the effects of Rb1 on high glucose-induced oxidative damage in rat RCECs and to investigate the underlying mechanism.

## 2. Results

### 2.1. Validation of Rat RCECs

Endothelial cells migrated from the retinal capillary fragments after 5 d (Figure 1a). In this study, cells acquired a typical contact-inhibited monolayer with a short fusiform or round morphology after 8 to 10 d (Figure 1b). The RCECs were analyzed for the expression and localization of endothelial cell markers by the immunofluorescence staining of CD31 and the von Willebrand factor (vWf). These two markers were also examined in positive control RCECs (Figure 1c,d).

### 2.2. Rb1 Increased Cell Viability in Rat RCECs Cultured in High Glucose Media

Cell viability was analyzed with MTT and trypan blue staining assays. In this study, we observed that incubation with 30 mM glucose for 48 h and 72 h significantly decreased the cell viability compared to incubation with 5.5 mM glucose media. As shown in Figure 2a, there was no significant difference in cell viability among the test groups after 24 h of incubation. We found that 10 μM and 20 μM of Rb1 increased the cell viability after 48 or 72 h of treatment (*p* < 0.05), and 5 μM of Rb1 increased the cell viability after 72 h of treatment (*p* < 0.05). As shown in Figure 2b, no significant differences were observed in the number of live cells after 24 h of treatment. Treatment with 20 μM Rb1 significantly increased the live cell count after 48 h of treatment (*p* < 0.05). Furthermore, we found that 10 μM and 20 μM of Rb1 significantly increased the live cell numbers after 72 h of treatment (*p* < 0.05). These results suggest that Rb1 can increase the cell viability in RCECs cultured in high glucose conditions.

### 2.3. Rb1 Ameliorated High Glucose-Induced Cytotoxicity, DNA Damage, and Apoptosis in Rat RCECs

The cell cytotoxicity level was determined with a lactate dehydrogenase (LDH) release assay, and the relative mtDNA copy number was measured in triplicate with a quantitative real-time PCR assay. The morphology of apoptotic cells was observed using Annexin V/PI double staining and laser confocal scanning microscopy. The fluorescence in cell clusters showed that apoptotic cells exhibited strong green and red fluorescence, while living cells exhibited little fluorescence. In this study, cellular LDH release was markedly increased and the mtDNA copy number was significantly decreased in the cells treated with 30 mM glucose compared with the cells treated with 5.5 mM glucose (*p* < 0.01). After culture for 72 h, Rb1 treatment significantly decreased the cellular LDH release (Figure 3a) and increased the mtDNA content (Figure 3b) in rat RCECs (*p* < 0.05). As shown in Figure 3c, cells grown in 5.5 mM glucose media showed little fluorescence. Exposure of RCECs to high glucose concentrations for 72 h increased the intensity of the green and red fluorescence. A markedly lower intensity of fluorescence was observed after treating RCECs with Rb1 (20 μM). These results suggest that Rb1 ameliorates high glucose-induced endothelial damage in rat RCECs.

### 2.4. Rb1 Attenuated High Glucose-Induced ROS Generation in Rat RCECs

ROS, particularly mtROS, are key players in the pathogenesis of diabetic vascular complications. In this study, we monitored intracellular ROS production with Amplite^TM^ ROS Red and mitochondrial ROS generation with MitoTracker Red CMH-XRos, along with fluorescence microscopy. As shown in Figure 4, a significant increase in ROS production was observed in the cells cultured in 30 mM glucose compared with the cells cultured in 5.5 mM glucose (*p* < 0.01). Treatment with 20 µM Rb1 significantly reduced the ROS content in RCECs (*p* < 0.01). These results suggest that Rb1 reduces high glucose-induced oxidative stress in RCECs.

### 2.5. Rb1 Elevated Antioxidant Enzyme Activity and Inhibited the Activity of NOX in Rat RCECs Exposed to High Glucose 

Superoxide dismutase (SOD) and catalase (CAT) provide the first line of defense against ROS in cells. A major source of ROS in endothelial cells is NOX, which catalyzes the production of superoxide from oxygen and NADPH. As shown in Figure 5a,b, when cells were stimulated with 30 mM glucose, significant decreases in the activity of SOD and CAT were observed (*p* < 0.01). In addition, NOX activity was significantly increased in RCECs following exposure to 30 mM glucose (Figure 5c) (*p* < 0.01). Treatment with 20 μM Rb1 significantly elevated the activity of SOD and CAT, but decreased NOX activity in rat RCECs (Figure 5a–c) (*p* < 0.01). These data demonstrate that Rb1 shows significant antioxidative effects against high glucose-induced oxidative stress in RCECs.

### 2.6. Rb1 Restored NAD^+^ Levels and Induced NAD-PARP-SIRT Signaling in RCECs Exposed to High Glucose

High glucose concentrations activated PARP, leading to the depletion of NAD^+^ and the inhibition of SIRT activity and expression (the NAD-PARP-SIRT pathway) [11,26]. PARP, a DNA repair enzyme, is activated by DNA breaks induced by oxidative stress. By consuming NAD^+^ to catalyze the poly(ADP-ribosyl)ation of acceptor proteins, PARP activation can lead to ATP depletion and cell death [27]. In mammals, SIRT1 is mainly localized to the nucleus and plays diverse physiological roles in cellular signaling and transcriptional regulation [28]. SIRT3 is localized to the mitochondria, where it deacetylates and activates a number of enzymes involved in modulating the mitochondrial function and ROS production [29]. In this study, PARP activity significantly increased in RCECs following exposure to 30 mM glucose (Figure 6a) (*p* < 0.01). In RCECs, the high glucose-induced depletion of NAD^+^ levels and decrease in the NAD^+^/NADH ratio (Figure 6b–d) were associated with reduced intracellular SIRT activities (Figure 6e) (*p* < 0.01). Consistently, the protein expression and mRNA expression of SIRT1/SIRT3 were markedly decreased in rat RCECs (Figure 6f–h) (*p* < 0.01). These abnormalities were prevented by Rb1, and treatment with Rb1 (20 μM) in RCECs under high glucose conditions reduced PARP activation (Figure 6a) and increased NAD^+^ availability (Figure 6b–d) (*p* < 0.05). Rb1 also increased SIRT activity and modulated the expression of SIRT1/SIRT3 (Figure 6e–h) (*p* < 0.05). These data suggest that Rb1 may increase SIRT activity and expression by blocking PARP activation and restoring NAD^+^ levels.

### 2.7. Rb1 Treatment Showed No Effect on NMNAT1 Expression in Rat RCECs under High Glucose Conditions

NMNAT1 is a nuclear enzyme in all organisms that is essential for intracellular NAD^+^ synthesis in the salvage/recycling pathway. As shown in Figure 7, high glucose decreased the NMNAT1 protein content and mRNA expression (*p* < 0.05), but treatment with 20 μM Rb1 showed no effect on NMNAT1 protein and mRNA expression in RCECs.

### 2.8. Rb1 Regulated the Intracellular Redox State in High Glucose Conditions

NADPH and GSH provide reducing power for antioxidant defense in cells. To investigate the redox status of the cells, we measured NADPH, NADP^+^, GSH, and GSSG levels. As shown in Figure 8, high glucose induced a significant decrease in NADPH and GSH concentrations and an increase in NADP^+^ and GSSG levels, causing the NADPH/NADP^+^ and GSH/GSSG ratios to significantly decrease (*p* < 0.01). RCECs treated with 20 μM Rb1 showed a significant increase in NADPH and GSH levels (*p* < 0.05), while the NADP^+^ and GSSG levels in these cells were markedly decreased (*p* < 0.05); consequently, the intracellular NADPH/NADP^+^ and GSH/GSSG ratios were increased (*p* < 0.01) (Figure 8). These data show that Rb1 maintains the intracellular redox balance by increasing the NADPH and GSH levels and reducing intracellular oxidative stress.

## 3. Discussion

In this study, we found that Rb1 significantly inhibited high glucose-induced oxidative stress, DNA damage, and cell apoptosis in cultured RCECs. Treatment with Rb1 reduced the loss of cell viability, improved the antioxidant capacity, and reduced the oxidative damage in RCECs. Interestingly, high glucose decreased the NMNAT1 protein content and mRNA expression, but treatment with Rb1 showed no effect on NMNAT1 expression in RCECs. We also showed that the protective effect of Rb1 appears to be mediated by the regulation of the NAD-PARP-SIRT signaling pathway and the maintenance of cellular redox homeostasis. This finding further supports the concept that Rb1 may have potential as a retinoprotective drug.

Oxidative stress contributes to DR pathogenesis [30], ROS-mediated damage to mtDNA, and an altered mitochondrial function [31]. The apoptosis of retinal capillary cells is an early event in the pathogenesis of DR, and oxidative damage to DNA is considered a key factor in DR [32,33]. Our results showed that after being cultured in 30 mM glucose medium for 72 h, the cell viability and live cell count were significantly decreased, while LDH release was increased. Coincubation with Rb1 significantly inhibited high glucose-induced cytotoxicity. Further research found that the exposure of RCECs to a high glucose concentration for 72 h led to a dramatic increase in ROS production, a reduced mtDNA copy number, and induced apoptosis. However, treatment with Rb1 together with 30 mM glucose markedly inhibited ROS formation and diminished mtDNA damage and cell apoptosis in RCECs. These data suggest that Rb1 has ameliorative effects on high glucose-induced endothelial damage in rat RCECs.

Cells are in a stable state known as redox homeostasis under normal physiological conditions. Redox homeostasis is maintained by the balance between continuous ROS generation and several mechanisms involved in antioxidant activity [34]. An overwhelming production of ROS leads to a prooxidant state also known as oxidative stress, which is a leading factor in the pathogenesis of diabetic complications [35,36]. NOX is a major contributor to ROS generation in endothelial cells [37]. SOD and CAT are two important scavenging enzymes that remove toxic free radicals. Our study revealed that intracellular ROS and mitochondrial ROS were significantly increased when RCECs were cultured in 30 mM glucose media. Treatment with Rb1 reduced high glucose-induced ROS production. The present study also showed that NOX activity increased and SOD and CAT activity markedly decreased in RCECs when they were exposed to high glucose. Rb1-treated RCECs showed decreased NOX activity and increased SOD and CAT activity. These data suggest that high glucose-induced ROS production exceeds the natural antioxidant capacity and leads to oxidative stress, and that Rb1 may exert its protective effect on RCECs by reducing ROS accumulation and increasing antioxidant enzyme activity.

NAD^+^ plays a critical role in various biological functions, and increasing NAD^+^ levels may present a new and exciting approach for preventing the decline in cellular energy and function in a number of diseases, including DM [38,39]. PARP mediates a reversible posttranslational modification of proteins involved in DNA repair. However, at high levels of oxidative damage, the extensive activation of PARP can rapidly lead to NAD^+^ failure and cell death [40]. Sirtuins, which belong to the class III histone deacetylase family and exhibit NAD-dependent protein lysine deacylase activity, have been identified as pivotal regulators of various cellular processes [41]. NMNAT1 and nicotinamide phosphoribosyltransferase (NAMPT) form a nuclear NAD^+^ salvage pathway that supplies NAD^+^ as a substrate for a variety of NAD^+^-dependent enzymes, including the protein-deacetylating sirtuins and PARP. Several studies have demonstrated that cellular NAD^+^ depletion and declines in sirtuin activity play critical roles in PARP-mediated cell death [42,43]. Here, our study shows that high glucose-induced depletion of NAD^+^ levels and decreases in the NAD^+^/NADH ratio are associated with reduced intracellular SIRT activity/expression. Consistently, intracellular PARP activity was significantly increased in the high glucose media. Treatment with Rb1 reduced PARP activation and increased NAD^+^ availability and SIRT activity/expression. However, Rb1 showed no effect on the protein or mRNA expression of the NAD biosynthetic enzyme NMNAT1 in RCECs. It has been demonstrated that NAMPT plays a role in the regulation of NAD/SIRT1 biological activity in obesity and insulin resistance [44]. Therefore, further studies are necessary to elucidate the mechanisms by which the NAMPT/NAD/SIRT signaling pathway is modulated by Rb1 in RCECs under a high glucose condition. These data suggest that Rb1 can protect against high glucose-induced injury by regulating the NAD-PARP-SIRT signaling pathway in RCECs.

Nuclear factor E2-related factor (NRF2) has a crucial role in the cellular antioxidant system. Recent studies have reported that SIRT1 enhances the activity of NRF2 and upregulates the expression of NRF2 downstream genes, such as NADPH quinineoxidoreductase-1 (NQO-1), hemeoxygenase 1 (HO-1), and SOD [45,46,47]. Other studies have found that glutamine plays a significant role in several essential metabolic processes [48,49]. In diabetes and its complications, the over-release of glutamate is associated with increased oxidative stress and a reduced mitochondrial function. Additionally, improved mitochondrial glutamate metabolism is possibly related to the upregulation of the SIRT1-PGC-1a pathway [50]. However, the molecular mechanism and role of the NAD/SIRT1/NRF2 signaling pathway in DR remain largely unexplored. Whether NAD/SIRT1 regulates glutamine metabolism in DR remains unclear. Given the recent progress, it would be intriguing to see if SIRT1/NAD/NRF2 signaling can be therapeutically used to relieve diabetic retinal damage, and how the activity of glutamine synthetase and metabolism would change.

Finally, the NADPH/NADP^+^ and GSH/GSSG ratios were calculated to monitor the redox state of the cells. NADPH is a coenzyme widely involved in a broad range of oxidation reduction reactions, which are important for the maintenance of reduced GSH levels and thus form an important endogenous antioxidant mechanism. NADPH is mostly generated from the pentose phosphate pathway by glucose-6-phosphate dehydrogenase (G6PDH) and 6-phosphogluconate dehydrogenase (6PGDH). High glucose-induced activation of the polyol pathway, which is catalyzed by aldose reductase (AR), converts glucose to sorbitol with the concomitant oxidation of NADPH to NADP^+^. The depletion of intracellular NADPH and GSH promotes NOX activity, which has emerged as a major source of ROS in endothelial cells. The cellular reducing power, as measured by the NADPH/NADP^+^ and GSH/GSSG concentration ratios, was significantly lower under high glucose conditions than under control conditions (5.5 mM glucose). Treatment with Rb1 markedly enhanced the levels of NADPH and GSH and increased the GSH/GSSG and NADPH/NADP^+^ ratios. When administered under conditions of high glucose, Rb1 could increase the reducing power in RCECs and exert significant antioxidative effects against high glucose-induced endothelial damage.

## 4. Materials and Methods

### 4.1. Reagents and Animals

Rb1 (Figure 1, purity >98%) was purchased from Chengdu Must Bio-Technology Co., Ltd. (Chengdu, China). The molecular formula for Rb1 is C_54_H_92_O_23_, and the molecular weight is 1109.26 g/mol. The chemical structure is shown in Figure 9. Male Sprague-Dawley rats weighing 90–110 g were purchased from Vital River Laboratory Animal Technology Co., Ltd. in Beijing, China. The certificate number was SCXK (Jing) 2012-0001. Animal care and experiments were carried out according to the Guide for the Care and Use of Laboratory Animals released by the National Institutes of Health (NIH publication No 85-23, revised 1985). All procedures involving animals were approved by the Animal Ethics Committee of Beijing University of Chinese Medicine (NO. BUCM-4-2017122501-4037, 25 December 2017). During the study, all efforts were made to minimize animal suffering and to reduce the number of rats used to the minimum necessary.

### 4.2. Cell Culture and Identification

Using aseptic techniques, rat eyes were enucleated and hemisected, and their lenses and vitreous humors were removed. The retinas were carefully removed, minced into small pieces, placed in phosphate buffer solusion (PBS, containing 5% penicillin/streptomycin, Corning Inc., Corning, NY, USA), and washed. The retinal fragments were incubated with 0.1% collagenase type II (Sigma Aldrich, St. Louis, MO, USA) at 37 °C for 30 min. Following harvesting, the retinal pieces were passed through a 70-cell strainer (Falcon, Lincoln Park, NJ, USA), collected by centrifugation at 174× g, and resuspended in endothelial cell growth medium (ECM, ScienCell, San Diego, CA, USA). The above suspension was transferred to a T-25 cm^2^ flask precoated with 1% gelatin (Thermo Fisher Scientific, Waltham, MA, USA) and cultured in a 5% CO_2_ humidified incubator at 37 °C. The medium was changed 48 h after plating and every 3 d thereafter. The adherent cells were grown until the cells were confluent. Then, the cells were dissociated with 0.25% trypsin (Thermo Fisher Scientific) in PBS at a 1:2 ratio. The RCECs were analyzed for the expression and localization of endothelial cell markers by the immunofluorescence staining of platelet endothelial cell adhesion molecule-1 (PECAM-1/CD31) and von Willebrand factor (vWf). Cultures were fixed in 4% paraformaldehyde and blocked with 10% normal goat serum (NGS) (Solarbio, Beijing, China). Cells were then incubated with primary antibodies (rabbit anti-CD31, 1:100; rabbit anti-vWf, 1:100; Santa Cruz Biotechnology Inc., Dallas, TX, USA) at 4 °C overnight. After incubation with the primary antibodies, a fluorescein-conjugated goat anti-rabbit secondary antibody (1:100, Zhong Shan Jin Qiao, Beijing, China) was used to visualize the signal, and the cells were stained with 4′,6-diamidino-2-phenylindole (DAPI, Cell Signaling Technology, Danvers, MA, USA). Fluorescent signals were scored using a fluorescence microscope with appropriate filters.

### 4.3. MTT and Trypan Blue Assays

The MTT assay is a widely accepted method used to assess cytotoxicity and cell viability [51]. For the MTT assay, RCECs were seeded in 1% gelatin-coated 96-well culture plates (Sigma Aldrich) at a density of 1 × 10^4^ cells per well and allowed to adhere overnight. Endothelial cell growth medium containing 5.5 mmol/L glucose was used as a control. The cells were incubated for 24 h, 48 h, or 72 h, with either 5.5 mM glucose or 30 mM glucose, with or without varying concentrations of Rb1 (5, 10, or 20 μM). MTT solution (Sigma-Aldrich, 10 μL, 5 mg/mL) was added to each well 4 h before termination. After 4 h of incubation, the cells were lysed, and the purple formazan crystals were solubilized with 150 μL DMSO (Sigma-Aldrich). The plate was analyzed on an enzyme-linked immunosorbent assay reader (Thermo LabSystems, OY, Helsinki, Finland). For the trypan blue assay, cells were seeded on 24-well culture plates at a density of 3 × 10^4^ cells per well. The cells were incubated for 24 h, 48 h, or 72 h, with either 5.5 mM glucose or 30 mM glucose, with or without varying concentrations of Rb1 (5, 10, or 20 μM). The cells were harvested and mixed with 0.4% trypan blue (Sigma-Aldrich) at a 1:1 ratio for exclusion staining. Living cells excluded the dye, while dead cells were stained blue. The cell number was then determined using a light microscope.

### 4.4. Lactate Dehydrogenase Release Assay

RCECs were seeded on 1% gelatin-coated 96-well culture plates (Sigma Aldrich) at a density of 1 × 10^4^ cells per well and allowed to adhere overnight before experimentation. After treatment with 20 μM Rb1 for 72 h, measurement of LDH activity was performed using a commercial cytotoxicity assay kit (Beyotime Biotechnology, Shanghai, China).

### 4.5. Measurement of MtDNA Copy Number

Total genomic DNA was isolated from RCECs, and the mtDNA copy number was determined using real-time PCR, as previously described, with some modifications [52]. DNA was isolated using a TIANamp Genomic DNA Kit (Tiangen Biotech, Beijing, China). The DNA purity was measured with a spectrophotometer (Thermo Fisher Scientific), and DNA samples with OD260/OD280 ratios in the range of 1.8–2.0 were considered for the experiments. The mtDNA copy number was measured using real-time PCR and was normalized to the expression of β-globin. The forward and reverse primers used to amplify mtDNA were 5′-CCTCCTAATAAGCGGCTCCT-3′ and 5′-GGAGCTCGATTTGTTTCTGC-3′, respectively. The forward and reverse primers used to amplify β-globin were 5′-TTGTGTTGACTCGCAACCTC-3′ and 5′-CTCAGCGCCAACATTATCAG-3′, respectively. All of the PCR primers were synthesized by Shengong Inc. (Shanghai, China). PCR was performed on a StepOnePlus Real-Time System (Applied Biosystems, Foster City, CA, USA) with SYBR Green Real Master Mix (Tiangen Biotech). The amplification conditions were as follows: initial denaturation at 95 °C for two minutes, followed by 35 cycles of denaturation at 95 °C for 30 s, annealing at 60 °C for one minute, and extension at 72 °C for one minute. The relative mtDNA copy number was calculated with the 2^−ΔΔCt^ method [53].

### 4.6. Cell Apoptosis Assay

To detect apoptosis or necrosis, cells were plated in chamber slides (3 × 10^4^ cells per well) (Corning Inc.), allowed to attach overnight, and exposed to various experimental conditions. After treatment for 72 h, the cells were washed with PBS and incubated at room temperature with Annexin V-FITC conjugates and propidium iodide, following the standard procedures for the Annexin V-FITC Apoptosis Detection Kit (Beyotime Biotechnology). After 20 min, apoptotic cells were detected with a Nikon A1R Eclipse Ti confocal microscope (Nikon Corporation, Tokyo, Japan).

### 4.7. Measurement of ROS Levels

Intracellular ROS levels were measured with a Fluorometric Intracellular ROS Activity Assay Kit (AAT Bioquest, Sunnyvale, CA, USA), according to the manufacturer’s instructions. Briefly, rat RCECs were plated onto gelatin-coated 96-well plates at a density of 2 × 10^4^ cells per well and were exposed to experimental conditions. After treatment with 20 μM Rb1 for 72 h, ROS generation was measured with Amplite ROS Red, and the fluorescence intensity was measured with a fluorescence plate reader (excitation 520 nM, emission 605 nM). MitoTracker Red CMH-XRos (Invitrogen Life Technologies Inc., Grand Island, NY, USA) was used to measure the production of mitochondrial ROS. After treatment for 72 h, cells were incubated with MitoTracker Red CM-H2XRos at a final concentration of 2.5 μM for 30 min at 37 °C. The cells were then washed with PBS and observed under a fluorescence microscope (Nikon Corporation). Confocal images were analyzed using Image-Pro Plus 6.0 software (Media Cybernetics, Silver Spring, MD, USA).

### 4.8. Measurement of the Activity of SOD and CAT

RCECs were grown on gelatin-coated six-well culture plates (Corning Inc.) and exposed to various experimental conditions for 72 h. SOD and CAT assay kits (Beyotime Biotechnology) were used for the evaluation of SOD and CAT activity, respectively. The protein concentrations in the samples were measured using a BCA (Bicinchoninic acid) Protein Assay Kit (Beyotime Biotechnology).

### 4.9. NOX Activity

NOX activity was measured as previously described [54]. In brief, cells were trypsinized, pelleted by centrifugation, and resuspended in cold Krebs-HEPES buffer (ICN Biomedicals Inc., Aurora, OH, USA). A cellular suspension (5 × 10^4^ cells) was then placed in a 96-well plate in a luminescence reader (Thermo Fisher Scientific), and dark-adapted lucigenin (10 μmol/L, Sigma-Aldrich) was added to start the reaction. Chemiluminescence was recorded every 15 s for 10 min. NADPH (final concentration: 100 μmol/L; Sigma-Aldrich) was added after measuring the background lucigenin chemiluminescence, with further measurements performed for another 10 min. The differences between the values obtained before and after adding NADPH were calculated, and the result represented the activity of NADPH oxidase.

### 4.10. PARP Activity

PARP activity was assayed using a Trevigen PARP Universal Chemiluminescent Assay Kit (Trevigen, Gaithersburg, MD, USA), according to the manufacturer’s instructions, with modifications. Briefly, the PARP enzyme activity of each sample (cell lysate) was estimated based on the incorporation of biotinylated poly(ADP-ribose) units onto histone proteins coated in a 96-well plate that was provided with the kit. The values were calculated from a standard curve generated using known amounts of PARP enzyme and were normalized to the protein content.

### 4.11. NAD^+^/NADH Assay

RCECs were seeded onto gelatin-coated six-well plates, allowed to adhere overnight, and exposed to experimental conditions. The NAD^+^ and NADH levels were measured using the NAD^+^/NADH Assay Kit (AAT Bioquest), according to the manufacturer’s instructions. After treatment with Rb1 for 72 h, cells (1 × 10^6^ per sample) were harvested and lysed in 100 μL of the lysis buffer provided with the kit. The intensity of the color of the reduced product, measured at 460 nm, was proportional to the NAD^+^ concentration in the sample. The NAD^+^/NADH ratio was then calculated from the calibration curves supplied with the kit.

### 4.12. SIRT Activity Assay

A SIRT activity assay was performed using a Universal SIRT Activity Assay Kit (Abcam, Cambridge, MA, USA), according to the manufacturer’s instructions. After treatment for 72 h, sample cells were collected, and nuclear fractions were prepared. SIRT activity was normalized to the total protein concentration in each sample. The values were determined on a microplate reader (Thermo Fisher Scientific) within 5 min at 450 nm. The absorbance values from cells exposed to the test compounds were normalized to the mean absorbance values of the control samples and are shown as the mean ± SD of three replicates.

### 4.13. Western Blot Analysis

The expression of cellular SIRT1, SIRT3, and NMNAT1 was determined by western blot assay. Rat RCECs were grown at a density of 1 × 10^6^ cells/cm^2^ in T-25 cm^2^ flasks and were then exposed to various experimental conditions. For the western blot assay, cells were prepared in lysis buffer (50 mM Tris [pH=7.5], 150 mM NaCl, 1% NP40, 0.5% sodium deoxycholate, 1 mM EDTA, and 0.1% SDS), and the protein concentrations were determined with a BCA protein assay. Equal amounts of the whole protein extract (8 μg) underwent electrophoresis on a 10% SDS-polyacrylamide gel and were transferred to a polyvinylidene difluoride membrane (Immobilon P; Millipore™, Billerica, MA, USA). After transfer, the membranes were incubated overnight at 4 °C with antibodies against NMNAT1 (Bioss Inc., Beijing, China), SIRT1 (Affinity Biosciences, Cincinnati, OH, USA), SIRT3 (ProteinTech Group, Chicago, IL, USA), and GAPDH (Cell Signaling Technology, Danvers, MA, USA). After washing, the membranes were incubated with horseradish peroxidase-conjugated secondary antibodies and visualized using enhanced chemiluminescence western blotting detection reagents. The relative densities of the bands were determined by image analysis with Image-Pro Plus 6.0 software.

### 4.14. Real-Time Quantitative Polymerase Chain Reaction (PCR) Analysis

The cells were harvested, and RNA was extracted using an Ultrapure RNA Kit (CWBIO, Beijing, China). cDNA was synthesized from 2 μg of total RNA using an HiFi-MMLV cDNA Kit (CWBIO). Real-time PCR was performed with the StepOnePlus Real-Time PCR System (Applied Biosystems) using an SYBR Green Master Mix Kit (CWBIO). Primer sets for the rat genes NMNAT1 (sense, 5′-GCAGACTTGCTGGAGTCCTT-3′; antisense, 5′-ATTTCTGAGCGTCACTGCCA-3′), SIRT1 (sense, 5′-TCTCCCAGATCCTCAAGCCA-3′; antisense, 5′-CTGCAACCTGCTCCAAGGTA-3′), SIRT3 (sense, 5′-GTGGCCTCTACAGCAACCTT-3′; antisense, 5′-AGGGCTTGGGGTTGTGAAAA-3′), and β-actin (sense, 5′-CCCATCTATGAGGGTTACGC-3′; antisense, 5′-TGCGGTGGACGATGGAGG-3′) were synthesized by Shengong Inc. The resulting PCR products were analyzed on a 1% agarose gel and with dissociation curves. To assess the amplification efficiency of the different pairs of primers, serial dilutions of control cDNA were amplified by real-time PCR using the primers for NMNAT1, SIRT1, SIRT3, and β-actin. The 2^−ΔΔCT^ method was used to analyze the relative changes in gene expression, and the expression of the transcripts is expressed relative to β-actin expression.

### 4.15. NADP^+^/NADPH and GSH/GSSG Assays

An NADP^+^/NADPH assay was performed using an NADP^+^/NADPH Ratiometric Kit (Promega, Madison, WI, USA), according to the manufacturer’s instructions. After treatment for 72 h, cell extracts (5 × 10^4^ per sample) were treated with NADP^+^ and NADPH extraction buffers. Chemiluminescence was recorded with an EnSpire Multimode Plate Reader (PerkinElmer, Boston, MA, USA). A GSH/GSSG assay was performed using a Green Fluorometric Detection Kit (AAT Bioquest). After treatment for 72 h, cells (1 × 10^6^ per sample) were harvested and lysed in 100 μL of lysis buffer to perform the GSH/GSSG assay. Fluorescence was measured in a plate reader with an excitation/emission wavelength of 490/520 nm.

### 4.16. Statistical Analysis

The results are given as the means ± standard deviation. One-way analysis of variance followed by a Student–Newman–Keuls (SNK) test was performed to analyze multiple comparisons. Statistical analyses were performed using SPSS 18.0 software (SPSS Inc., Chicago, IL, USA). Significance was set at *p* < 0.05.

## 5. Conclusions

In conclusion, in this study, we identified a novel regulatory mechanism for NMNAT1, SIRT1, and SIRT3 reduction in high glucose conditions that involved PARP activation-induced increases in ROS production through NOX activation and a reduction in the antioxidation ability, leading to the depletion of NAD^+^ and disturbed intracellular redox homeostasis. The protective effects of Rb1 are related to the regulation of the NAD-PARP-SIRT signaling pathway and maintenance of the cellular reducing power; this effect of Rb1 may help to provide new insights for developing novel treatments for DR. However, further studies are still needed to elucidate the exact mechanism by which the NAD(H)/NADP(H) metabolic pathways are activated by Rb1 treatment.

## Figures and Tables

**Figure 1 ijms-20-04936-f001:**
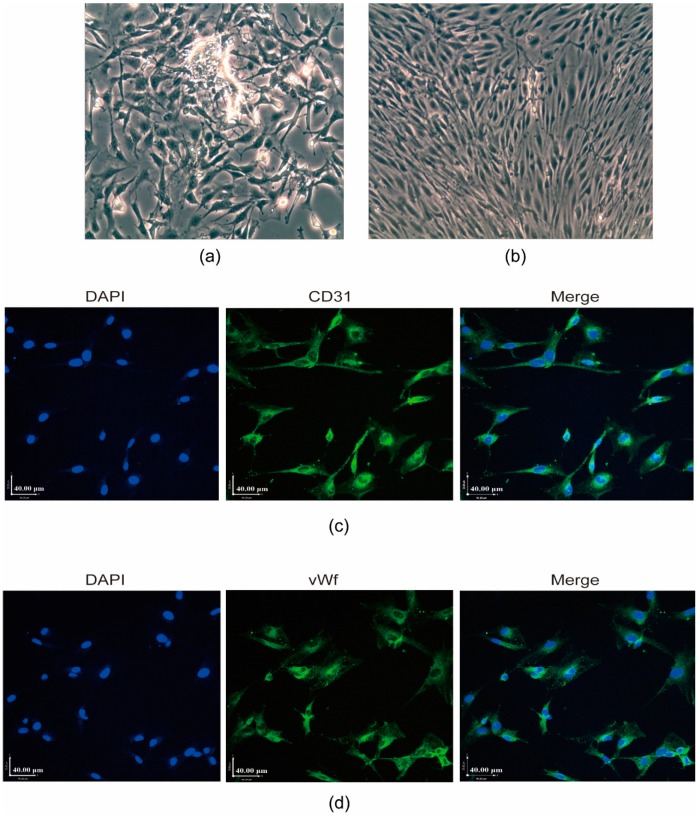
Characterization of established rat retinal capillary endothelial cell (RCEC) cultures. (**a**) Endothelial cells migrated from the retinal capillary fragments after 5 d. (**b**) Cells acquired a typical contact-inhibited monolayer with a short fusiform or round morphology after 8 to 10 d. The original magnification was 200×. (**c**) and (**d**) The RCECs were analyzed for the expression and localization of endothelial cell markers by immunofluorescence staining of CD31 and the von Willebrand factor (vWf). All nuclei were stained with 4′,6-diamidino-2-phenylindole (DAPI) (blue fluorescence). Scale bar: 40 μm.

**Figure 2 ijms-20-04936-f002:**
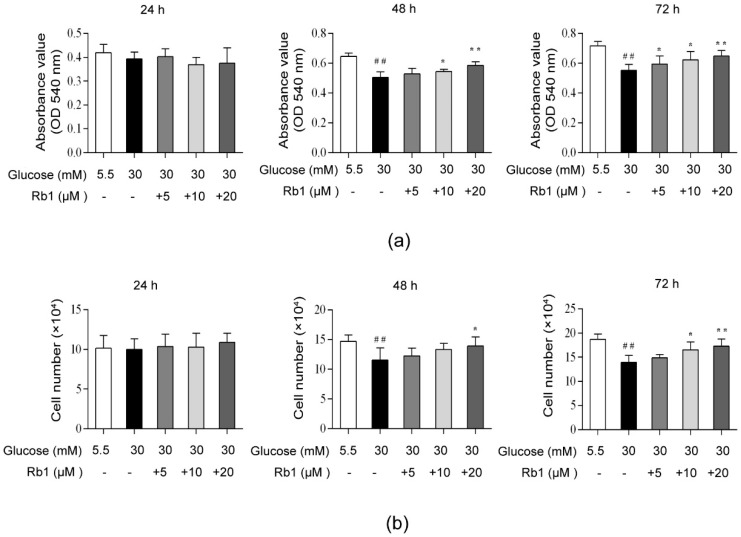
Ginsenoside Rb1 (Rb1) increased the cell viability in rat RCECs exposed to high glucose. (**a**) MTT assays were used to examine the cell viability. (**b**) Trypan blue exclusion was used to count the live cells. The data are expressed as the means ± SD. ## *p* < 0.01 versus 5.5 mM glucose; **p* < 0.05 and ** *p* < 0.01 versus 30 mM glucose.

**Figure 3 ijms-20-04936-f003:**
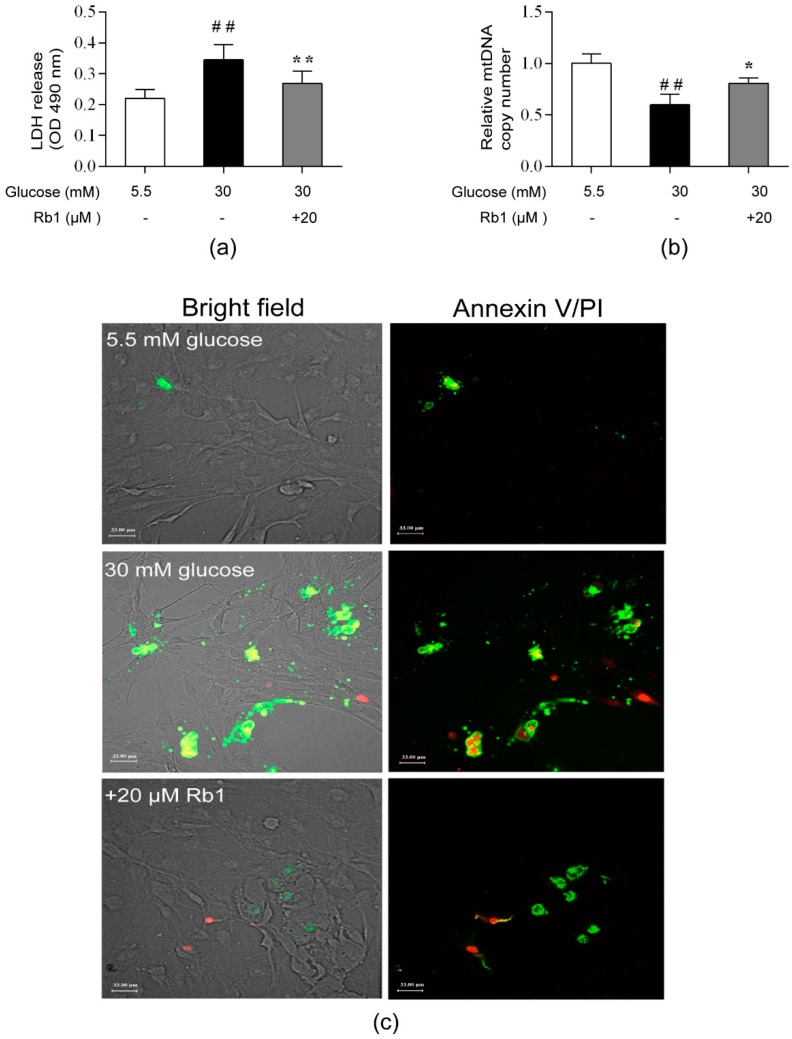
Rb1 ameliorated high glucose-induced cytotoxicity, DNA damage, and apoptosis in rat RCECs. (**a**) A lactate dehydrogenase (LDH) assay was used to assess the cell cytotoxicity. (**b**) The relative mtDNA copy number was estimated with real-time quantitative PCR. (**c**) Apoptosis was measured with an Annexin V-FITC/PI double staining assay and laser confocal scanning microscopy. The data are expressed as the means ± SD. ## *p* < 0.01 versus 5.5 mM glucose; * *p* < 0.05 and ** *p* < 0.01 versus 30 mM glucose. Scale bar: 33 μm.

**Figure 4 ijms-20-04936-f004:**
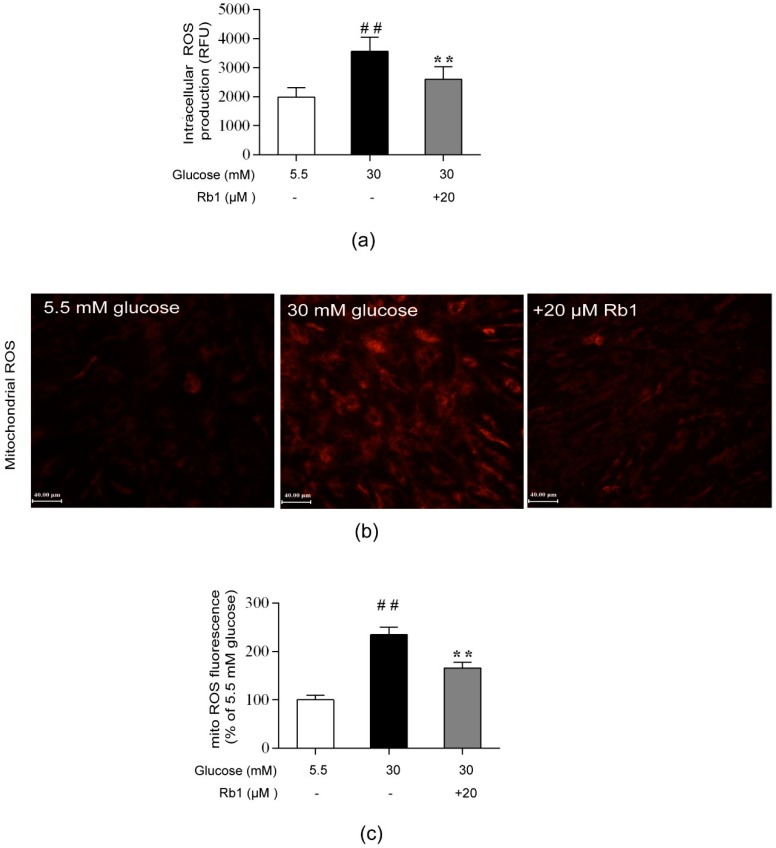
Rb1 inhibited ROS generation induced by high glucose in RCECs. (**a**) Intracellular ROS generation was measured with Amplite ROS Red. (**b**) A MitoTracker Red CMH-XRos probe was used to monitor ROS production in mitochondria. (**c**) Fluorescence images were analyzed using Image-Pro Plus 6.0 software. The data are expressed as the means ± SD. ## *p* < 0.01 versus 5.5 mM glucose; ** *p* < 0.01 versus 30 mM glucose. Scale bar: 40 μm.

**Figure 5 ijms-20-04936-f005:**
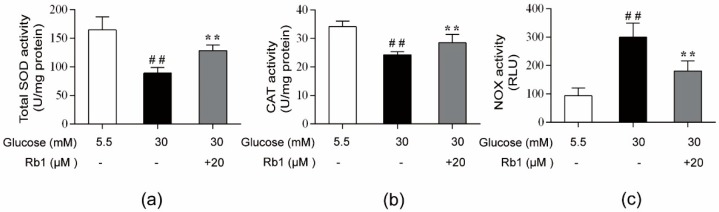
Effects of Rb1 on the activity of nicotinamide adenine dinucleotide phosphate (NADPH) oxidase (NOX) and antioxidant enzymes in RCECs exposed to high glucose. (**a**) and (**b**) The activity of superoxide dismutase (SOD) and catalase (CAT) was evaluated with an enzymatic colorimetric method. (**c**) The activity of NOX was measured with the chemiluminescence method. The data are expressed as the means ± SD. ## *p* < 0.01 versus 5.5 mM glucose; ** *p* < 0.01 versus 30 mM glucose.

**Figure 6 ijms-20-04936-f006:**
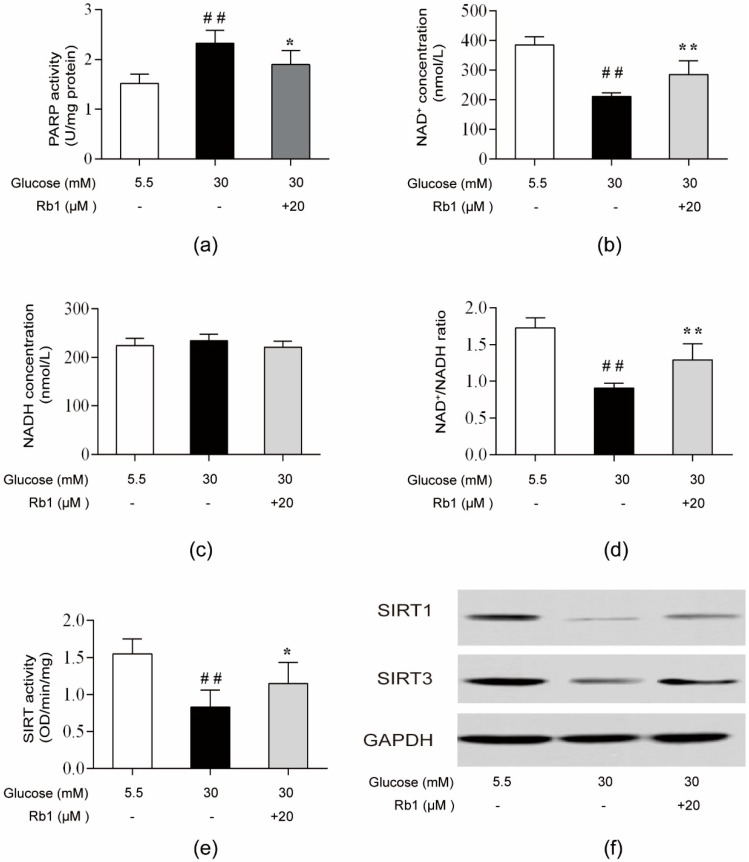
Rb1 prevented the reductions in sirtuin1 (SIRT1), SIRT3, and the nicotinamide adenine dinucleotide (NAD^+^)/NADH ratio and abrogated the increase in poly(ADP-ribose) polymerase (PARP) in RCECs exposed to high glucose. (**a**) Cells were collected to examine PARP activity. (**b**) NAD^+^ levels. (**c**) NADH levels. (**d**) Intracellular NAD^+^/NADH ratios. (**e**) Sirtuin activity. (**f**) Western blot analysis was used to determine SIRT1 and SIRT3 protein expression in RCECs. (**g**) The relative densities of SIRT1 and SIRT3 protein expression were determined by image analysis with Image-Pro Plus 6.0 software. (**h**) RT-qPCR was used to measure SIRT1 and SIRT3 mRNA expression. The data are expressed as the means ± SD. ## *p* < 0.01 versus 5.5 mM glucose; * *p* < 0.05 and ** *p* < 0.01 versus 30 mM glucose.

**Figure 7 ijms-20-04936-f007:**
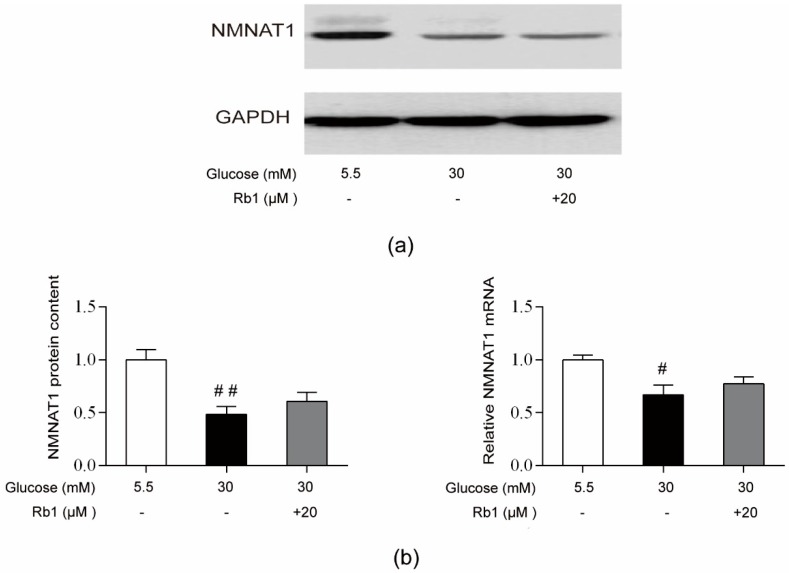
Rb1 treatment showed no effect on NMNAT1 expression in rat RCECs exposed to high glucose. (**a**) Western blot analysis was used to determine NMNAT1 protein expression. (**b**) The relative densities of NMNAT1 protein expression were determined by image analysis with Image-Pro Plus 6.0 software. (**c**) RT-qPCR was used to determine NMNAT1 mRNA expression. Data are expressed as the means ± SD. ## *p* < 0.01 versus 5.5 mM glucose; ** *p* < 0.01 versus 30 mM glucose.

**Figure 8 ijms-20-04936-f008:**
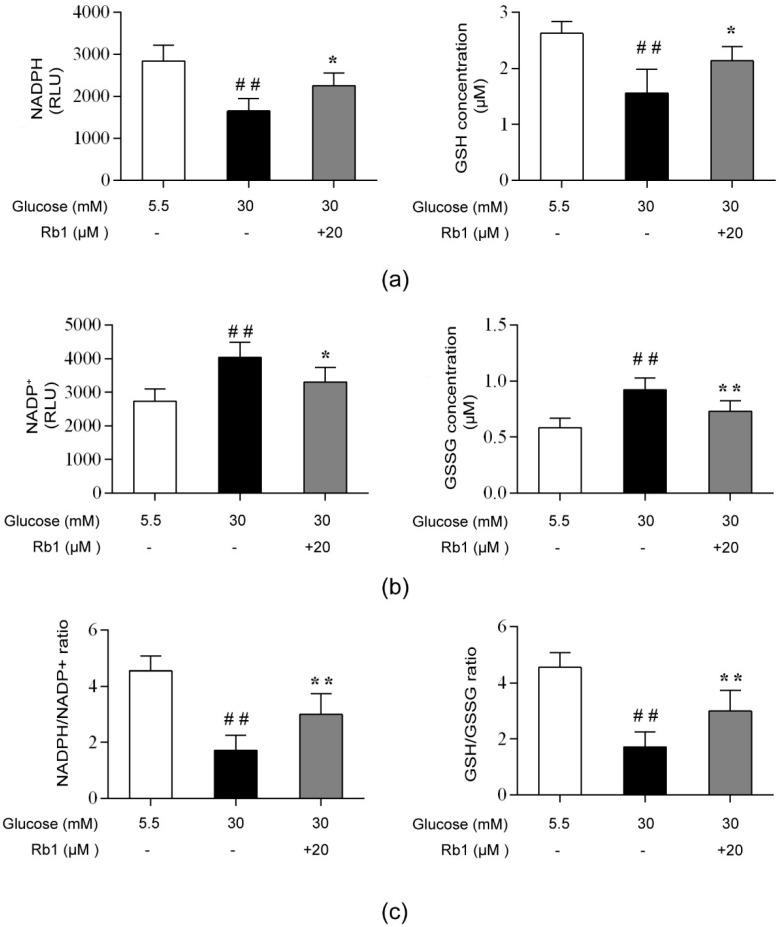
Rb1 modulated the NADPH/NADP^+^ and glutathione (GSH)/GSSG ratios in RCECs exposed to high glucose. (**a**) NADPH and GSH levels. (**b**) NADP^+^ and GSSG levels. (**c**) NADPH/NADP^+^ and GSH/GSSG ratios. Data are expressed as the means ± SD. ## *p* < 0.01 versus 5.5 mM glucose; * *p* < 0.05 and ** *p* < 0.01 versus 30 mM glucose.

**Figure 9 ijms-20-04936-f009:**
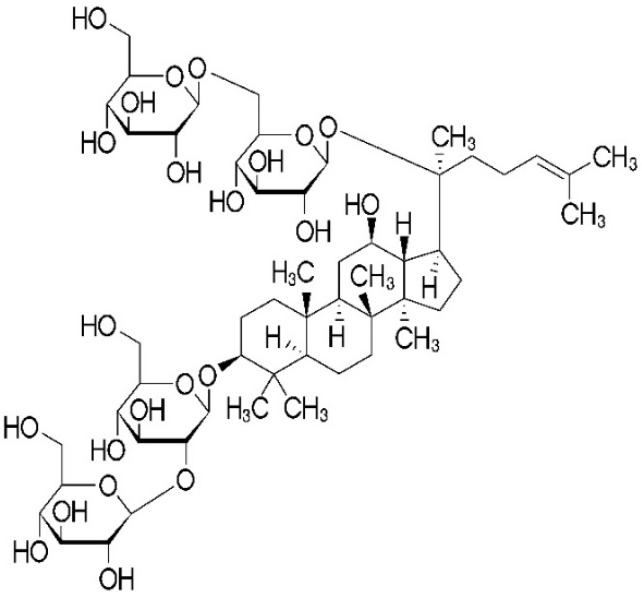
Chemical structure of Ginsenoside Rb1.

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
