# Peer review of "Ginsenoside Rb1 Attenuates High Glucose-Induced Oxidative Injury via the NAD-PARP-SIRT Axis in Rat Retinal Capillary Endothelial Cells"

_ijms, 2019, doi:10.3390/ijms20194936_

Round 1

Reviewer 1 Report

The paper of Fan and co-workers reports on the protective effects of ginsenoside Rb1 on high-glucose damage in retinal capillary endothelial cells. The injuries induced by high-glucose on cell viability and molecular mechanisms are attenuated or reverted through the activation of the NAD-PARP-SIRT axis.

The paper is interesting and results sound, however this referee has some concerns related to manuscript style.

The authors should follow the editorial policy for manuscript organization (subheadings without numbering, materials and methods after the introduction, etc..) and reference style. After that reference numbering should appropriately be checked. Please specify the abbreviation at first mention (i.e. NMAT1 is not explicated in the abstract) and then follow the same style in the rest of the manuscript (i.e. Nmnat1at line 50 should be NMNAT1). Please add NMNAT1 at line 402 in the proteins analyzed by WB. Figure 3: the bright filed images on the left should be really bright and clear. As they are now it seems they are merging of bright field and fluorescence. You can keep them as a third series of merged images on the right. Please specify in the legend of the third panel that “+20 uM Rb1” instead of “20 uM Rb1”.

Author Response

Dear Reviewer:

Thank you very much for your comments on the manuscript entitled “Ginsenoside Rb1 attenuates high glucose-induced oxidative injury via the NAD-PARP-SIRT axis in rat retinal capillary endothelial cells” (ijms-599977). Based on your comments, we attached a point-by-point answer. The manuscript was revised accordingly. Please refer to the answer listed below:

Point 1: The authors should follow the editorial policy for manuscript organization (subheadings without numbering, materials and methods after the introduction, etc.) and reference style. After that reference numbering should appropriately be checked.

Response 1: Thanks for your careful checks. We used the “IJMS Microsoft Word template” to prepare our manuscript, so the results section was put after the introduction. We also have carefully checked all the references in the manuscript.

Point 2: Please specify the abbreviation at first mention (i.e. NMAT1 is not explicated in the abstract) and then follow the same style in the rest of the manuscript (i.e. Nmnat1at line 50 should be NMNAT1). Please add NMNAT1 at line 402 in the proteins analyzed by WB.

Response 2: Thanks for your careful checks. The abbreviation of NMNAT1 was added in the abstract (line 23). “Nmnat1” at line 51 was changed to “NMNAT1”. And line 427, “and NMNAT1” was added in the proteins analyzed by WB.

Point 3: Figure 3: the bright filed images on the left should be really bright and clear. As they are now it seems they are merging of bright field and fluorescence. You can keep them as a third series of merged images on the right.

Response 3: The illuminator of microscope in our lab didn’t work properly, so the bright filed images were not clear enough. In order to keep track of the damaged cell, I merged images on the microscope software system.The images had already been stored, which were kept in their original format. We feel sorry for the images in Figure 3 that can’t be modified.

Point 4: Please specify in the legend of the third panel that “+20 uM Rb1” instead of “20 uM Rb1”.

Response 4: Thank you for your suggestion. We have improved all the legend of figures in the whole manuscript.

We tried our best to improve the manuscript and made some changes in the manuscript. These changes will not influence the content and framework of the paper. And here we marked via enabling the ‘Track changes’ feature of Microsoft Word in revised paper. We appreciate for Editors’ and Reviewers’ warm work earnestly, and hope that the correction will meet with approval.

Yours sincerely

Chunlan Fan

Reviewer 2 Report

Nice work by Dr. Tang and group describing the role of SIRT pathway under high glucose induced oxidative damage. Few things should be addressed before it is ready for acceptance. They are as follows:

Authors should mention their perspective on NRF2 antioxidant pathway in this context. It will be interesting to think about that perspective. They should mention few lines about this context in discussion part. MTT assay has been done to measure cell proliferation. There should be an argument why authors think MTT is an idea assay in this context as metabolism might play some role for false reading in case of MTT. MTT assay is a measurement of cell metabolic activity which determines the state of cell proliferation. NAD-SIRT regulates metabolism. So it should be argued why authors think MTT assay is an idea way to quantify the cell proliferation in this context. They can add few lines on this argument in discussion part as well. Another point must be noteworthy to see how glutamine metabolism  might play a role in this context. As mentioned in doi: 10.18632/oncoscience.253/ PMID: 26682255 glutamine plays a significant role in metabolic check points. So it will be intriguing to speculate a key role played by glutamine  metabolism in this context. By referring the above mentioned work authors should add few lines on this context. It will help to address the global broader role of SIRT/NAD pathway in metabolism context.

Author Response

Dear Reviewer:

Thank you very much for your comments on the manuscript entitled “Ginsenoside Rb1 attenuates high glucose-induced oxidative injury via the NAD-PARP-SIRT axis in rat retinal capillary endothelial cells” (ijms-599977). Based on your comments, we attached a point-by-point answer. The manuscript was revised accordingly. Please refer to the answer listed below:

Point 1: Authors should mention their perspective on NRF2 antioxidant pathway in this context. It will be interesting to think about that perspective. They should mention few lines about this context in discussion part.

Response 1: Thank you for your helpful suggestion. We have added “Nuclear factor E2-related factor (NRF2) has a crucial role in the cellular antioxidant system. Recent studies have reported that SIRT1 enhances the activity of NRF2 and upregulates the expression of NRF2 downstream genes, such as NADPH quinineoxidoreductase-1 (NQO-1), hemeoxygenase 1 (HO-1) and SOD.” at line 273-276. Page 11, line 280-282, “However, the molecular mechanism and role of the NAD/SIRT1/NRF2 signaling pathway in DR remains largely unexplored.” was added. Line 282-284, “Given the recent progress, it would be intriguing to see if SIRT1/NAD/NRF2 signaling can be therapeutically used to relieve diabetic retinal damage, and how the activity of glutamine synthetase and metabolism would change.” was added.

Point 2: MTT assay has been done to measure cell proliferation. There should be an argument why authors think MTT is an idea assay in this context as metabolism might play some role for false reading in case of MTT. MTT assay is a measurement of cell metabolic activity which determines the state of cell proliferation. NAD-SIRT regulates metabolism. So it should be argued why authors think MTT assay is an idea way to quantify the cell proliferation in this context. They can add few lines on this argument in discussion part as well.

Response 2: In the study, MTT assay is used to assess the cellular cytotoxicity and viability in high glucose condition. We have made the appropriate changes. (1) Line 333, the statements of “Cell Proliferation Assay” were changed to “MTT and Trypan Blue Assays”. (2) Line 334, the statements of “The proliferation of RCECs was determined using MTT and trypan blue staining assays” were changed to “The MTT assay was the widely accepted method to assess cytotoxicity and cell viability”.

Point 3: Another point must be noteworthy to see how glutamine metabolism might play a role in this context. As mentioned in doi:10.18632/oncoscience.253/PMID:26682255 glutamine plays a significant role in metabolic check points. So it will be intriguing to speculate a key role played by glutamine metabolism in this context. By referring the above mentioned work authors should add few lines on this context. It will help to address the global broader role of SIRT/NAD pathway in metabolism context.

Response 3: Thank you for your helpful suggestion. We make the following changes: (1) Line 276-280, “Other studies have found that glutamine plays a significant role in several essential metabolic processes. In diabetes and its complications, over-release of glutamate is associated with increased oxidative stress and reduced mitochondrial function. While improved mitochondrial glutamate metabolism possibly related to upregulation of the SIRT1-PGC-1a pathway.” Was added. (2) Line 281-284, “Whether NAD/SIRT1 regulates glutamine metabolism in DR remains unclear. Given the recent progress, it would be intriguing to see if SIRT1/NAD/NRF2 signaling can be therapeutically used to relieve diabetic retinal damage, and how the activity of glutamine synthetase and metabolism would change.” was added.

We tried our best to improve the manuscript and made some changes in the manuscript. These changes will not influence the content and framework of the paper. And here we marked via enabling the ‘Track changes’ feature of Microsoft Word in revised paper. We appreciate for Editors’ and Reviewers’ warm work earnestly, and hope that the correction will meet with approval.

Yours sincerely

Chunlan Fan